

# An analysis of the cloud environment over the Ross Sea and Ross Ice Shelf using CloudSat/CALIPSO satellite observations: The importance of synoptic forcing

Ben Jolly[1], Peter Kuma[2], Adrian McDonald[2], and Simon Parsons[2]

[1]Landcare Research, Lincoln, New Zealand.
[2]Department of Physics and Astronomy, University of Canterbury, New Zealand.

*Correspondence to:* Ben Jolly (jollyb@landcareresearch.co.nz)

**Abstract.** We use the 2B-GEOPROF-LIDAR R04 (2BGL4) and R05 (2BGL5) products and the 2B-CLDCLASSLIDAR R04 (2BCL4) product, all generated by combining CloudSat radar and CALIPSO lidar satellite measurements with auxiliary data, to examine the vertical distribution of cloud occurrence around the Ross Ice Shelf (RIS) and Ross Sea region. We find that the 2BGL4 product, used in previous studies in this region, displays a discontinuity at 8.2 km which is not observable in the other products. This artefact appears to correspond with a change in the horizontal and vertical resolution of the CALIPSO dataset used above this level. We then use the 2BCL4 product to examine the vertical distribution of cloud occurrence, phase, and type over the RIS and Ross Sea. In particular we examine how synoptic conditions in the region, derived using a previously developed synoptic classification, impact the cloud environment and the contrasting response in the two regions. We observe large differences between the cloud occurrence as a function of altitude for synoptic regimes relative to those for seasonal variations. A stronger variation in the occurrence of clear skies and multi-layer cloud and in all cloud type occurrences over both the Ross Sea and RIS is associated with synoptic type than seasonal composites. In addition, anomalies from the mean joint histogram of cloud top height against thickness display significant differences over the Ross Sea and RIS sectors as a function of synoptic regime, but are near identical over these two regions when a seasonal analysis is completed. However, the frequency of particular phases of cloud, notably mixed phase and water, is much more strongly modulated by seasonal than synoptic regime compositing which suggests that temperature is still the most important control on cloud phase in the region


# 1 Introduction

Introduction

Antarctic tropospheric clouds have been the subject of many studies, including relevant reviews
by Lachlan-Cope (2010) and Bromwich et al. (2012). Detailed ground or airborne observation cam-
paigns (e.g. Scott and Lubin (2014, 2016)) are difficult, expensive to conduct and rare in this re-
gion (Lachlan-Cope, 2010), however satellite measurements have made a number of useful insights
possible (Verlinden et al., 2011; Bromwich et al., 2012; Adhikari et al., 2012). The properties of
snow- and ice-covered ground – namely being white, highly reflective, and very cold – pose chal-
lenges to the use of passive satellite sensors for cloud identification (Frey et al., 2008). These chal-
lenges are largely circumvented by the active instruments on the CloudSat (Stephens et al., 2008)
and CALIPSO (Winker et al., 2009) satellites whose data we use in this study. While detailed at-
mospheric models potentially allow further studies over far greater regional and temporal scales
(Fogt and Bromwich, 2008; Nicolas and Bromwich, 2011; Steinhoff et al., 2009), cloud is difficult
to model and accurately forecast, particularly over Antarctica and the Southern Ocean (Bromwich
et al., 2012), with the paucity of observations a contributing factor.

The Antarctic coastal region is one of the most active areas of synoptic-scale cyclonic storms in
the Southern Hemisphere (Hoskins and Hodges, 2005), with Adhikari et al. (2012) suggesting that
these lows are associated with deep and high-level clouds and precipitation. Additionally, Tsukernik
and Lynch (2013) identified that the meridional moisture flux is dominated by motions at synop-
tic scales and reveal that the Amundsen Sea sector experiences the highest variability around the
Antarctic, a potential driver of the variability observed in the region. This study focuses on cloud
properties over the Ross Sea and the Ross Ice Shelf (RIS) because these regions are of particular in-
terest in understanding the controls of cloud properties around Antarctica. For example, it has been
reported that the largest seasonal variations in cloud occurrence across the Antarctic are observed
in these regions with close to 60% during winter and 90% in the summer (Adhikari et al., 2012). A
number of recent studies (Scott and Lubin, 2014, 2016) have also identified unique cloud properties
in these regions and case studies detailed in Scott and Lubin (2014) suggest a strong dependence on
meteorological scenario.

The RIS is a largely flat expanse of permanent ice fed by both the West Antarctic Ice Sheet (WAIS)
and East Antarctic Ice Sheet (EAIS). The western edge of the shelf is bounded by the 2 km high
barrier of the Transantarctic Mountains (TAM), with the EAIS behind. The surface meteorology of
the region is dominated by katabatic winds from the ice sheets (Parish and Bromwich, 1991, 2007),
and low-pressure systems over the Ross Sea. The Ross Sea is located along the northern boundary
of the RIS and frequently experiences large low-pressure systems originating off the coast of Adélie
Land located well to the north-west. These are known to advect moist marine air from the ocean/sea
ice onto the RIS, often via the WAIS and Siple Coast (Nicolas and Bromwich, 2011). Nicolas and
Bromwich (2011) also highlighted the importance of marine air intrusions on cloud fraction over





the West Antarctic Ice Sheet driven by cyclonic activity in the Ross and western Amundsen Sea.
This combination of cyclones, the barrier presented by the TAM, and katabatic drainage helps to

feed a southerly wind regime that dominates the climatology of the RIS known as the Ross Ice
Shelf airstream (RAS) (Parish et al., 2006). Steinhoff et al. (2009) discussed a case study where a
cyclone off Marie Byrd Land transported moisture across the WAIS to the southern base of the RIS
which formed into cloud due to both low-level convergence and lifting caused by a "knob flow". A
distinct extended thermal infrared signature hypothesised to be associated with low-level cloud was

observed along the corridor of high winds linked to this RAS event. Recent work by Coggins et al.
(2014) has developed a synoptic classification scheme by applying the *k*-means clustering method
to 33 years of ERA-Interim surface wind data. This has been useful in understanding the range,
frequency, and influence of the different phenomena around the RIS (see Section 2.2 for details).
More recent work by Coggins and McDonald (2015) demonstrated how the position and depth of

the Amundsen Sea Low influences the frequency and form of these different weather regimes over
the Ross Sea and RIS. This study aims to quantify cloud occurrence over the RIS and southern
Ross Sea using the CloudSat/CALIPSO 2B-CLDCLASS-LIDAR product (Sassen et al., 2008), both
spatially and vertically. We also examine the occurrence, phase, and type of cloud with a focus on
whether synoptic drivers, identified via the synoptic regimes developed by Coggins et al. (2014),

provide a coherent pattern.

Clouds over the Southern Ocean and Antarctica can consist of liquid water, mixed-phases (i.e.
consisting of supercooled liquid water droplets and ice crystals), or ice crystals (Haynes et al., 2011;
Chubb et al., 2013; Scott and Lubin, 2014; Lawson and Gettelman, 2014). Cloud phase is important
to determine because ice crystals and water droplets have different radiative properties and therefore

reflect and absorb different levels of incoming shortwave radiation (Haynes et al., 2011; Scott and
Lubin, 2014). Cloud composition over Antarctica and the Southern Ocean is currently not well understood or modelled, however Lawson and Gettelman (2014) have shown that the radiative budget
in this area is highly sensitive to changes in cloud phase.

Chen et al. (2000) have shown that different types of clouds have distinctive microphysical properties, resulting in different radiative forcings (Chen et al., 2000; Tselioudis et al., 2013; Oreopoulos

et al., 2016; McDonald et al., 2016). It is therefore clear that classification is an important task.
The International Satellite Cloud Climatology Project (ISCCP) uses passive measurements to classify clouds into nine different types based on their cloud top pressure and cloud optical thickness
(Rossow and Schiffer, 1999). Later work by Wang and Sassen (2001) developed an approach to clas-

sify clouds into eight types by combining radiometer observations with "active" measurements from
ground-based lidar and radar. This classification scheme was modified for CloudSat and CALIPSO
observations to provide cloud type distributions globally which are available in the 2B-CLDCLASS-
LIDAR R04 (2BCL4) product used in this study (Sassen et al., 2008; Wang, 2012).




A recent study by Scott and Lubin (2014) investigated clouds over McMurdo station, located at
the north-west corner of the RIS, using spectroradiometer measurements as well as observations
from the NASA A-Train satellites. They identified two major sources of moisture: marine air intru-
sions originating over the WAIS which then cross the RIS (predominantly ice-based), and moist air
advection from the Ross Sea (more likely to contain liquid). Large cyclones in the Ross Sea did not
contribute significant levels of moisture at Ross Island. In a follow-up study, Scott and Lubin (2016)
extended this work to show a link between high ice content and increased vertical motion of the air
parcel prior to observation.

Verlinden et al. (2011) used vertical profiles of cloud occurrence from a pre-R05 2B-GEOPROF-
LIDAR product (Mace et al., 2009; Mace and Zhang, 2014). They found a pronounced seasonal
cycle in cloudiness over Antarctica and the Southern Ocean with higher cloud occurrences during
the winter. They also found a nearly discontinuous drop-off in cloudiness near 8 km over much of
the continent. However, they and the review by Bromwich et al. (2012) have questioned whether this
is an artefact in the data because this discontinuity corresponds with a change in the horizontal and
vertical resolutions of the CALIPSO data. Verlinden et al. (2011) also highlighted that their vertical
profiles revealed two distinct maxima with one near the surface level and the other near the top of
the troposphere.

The increase of cloud during winter is contrary to the findings of Adhikari et al. (2012), who
calculated seasonal variations spatially and found that summer and autumn featured higher cloud
occurrence than winter and spring over most of Antarctica and the Southern Ocean, but particularly
over the RIS. Sea ice was suggested as a contributing factor, blocking evaporation that occurs over
open water, along with the extremely low temperatures. Low-level cloud featured the highest inter-
seasonal variability, with low occurrence during winter and reduced occurrence during spring relative
to summer and autumn. Haynes et al. (2011) also examined clouds over the Southern Ocean using a
combination of active and passive satellite data. They separated the clouds in this region into eight
regimes, but identified that all of these regimes contained a relatively high occurrence of low cloud,
with 79% of all cloud layers observed featuring tops below 3 km in altitude. Multi-layered cloud
systems were observed in approximately 34% of cloud profiles. Haynes et al. (2011) also found that
cloud systems are geometrically thicker during the austral winter and that all of the eight regimes
show enhanced low-level cloud fraction in the summer but that the seasonal variation at higher levels
is more complex. Those regimes found to be most closely associated with mid-latitude cyclones also
produced precipitation more frequently.





## 2 Datasets and Methods

### 2.1 CloudSat/CALIPSO data

CloudSat (Stephens et al., 2008) and CALIPSO (Winker et al., 2009) are two satellites that exist within the NASA "A-Train", a constellation of satellites with identical orbits that pass over the same parts of the earth within a narrow time window (less than 1 km apart 90% for the time period used in this study (Mace and Zhang, 2014). CloudSat carries a millimeter-wavelength (94 GHz) cloud profiling radar (CPR) with a vertical resolution of 240 m and a sea-level footprint of 1.4 km × 1.7 km. It detects tiny water droplets within clouds while also penetrating through optically dense upper layers to detect further layers at lower altitudes, however studies have shown that it struggles to resolve cloud below 1 km above ground level due to ground clutter (Mace et al., 2009). The Cloud-Aerosol Lidar with Orthogonal Polarization (CALIOP) instrument carried by the CALIPSO satellite provides vertical resolution of the order of 30 to 60 m with a roughly circular sea-level footprint 100 m in diameter. It is able to accurately detect cloud down to ground level, but has reduced sensitivity during daylight operations and cannot penetrate thick cloud. In particular, this study uses the 2B-GEOPROF-LIDAR R04 (2BGL4) and R05 (2BGL5) products (Mace et al., 2009; Mace and Zhang, 2014) which combine the CALIOP and CPR observations to examine the vertical distribution of cloud occurrence. We also use the 2BCL4 product which provides cloud occurrence, phase, and cloud type information using a combination of CPR, CALIOP, and MODIS output with ancillary temperature information from the European Centre for Medium-Range Weather Forecasts (ECMWF). Analysis presented in Section 3 shows that the pre-R04 2B-GEOPROF-LIDAR products (Verlinden et al., 2011; Adhikari et al., 2012; Bromwich et al., 2012, amongst others) display a discontinuity at 8.2 km which appears to be limited to the poles in both regions. Mace et al. (2009) indicates that the CALIOP data have a centre to centre pacing of 333 m between profiles in the horizontal and a 30 m vertical resolution below 8.2 km. Above 8.2 km, further averaging is applied to create a 1 km along-track resolution and a 60 m resolution in the vertical. Thus, we believe that the observed discontinuity is related to this change. We therefore focus our analysis on the use of the 2BCL4 product.

The 2BCL4 product classifies clouds by examining the vertical profiles and horizontal extent of clouds derived from the CPR and CALIOP measurements, the presence of precipitation, cloud temperature from ancillary ECMWF predictions, and upward radiances from MODIS measurements (Wang, 2012) and is consistent with the previous ISCCP classification (Rossow and Schiffer, 1999). The clustering algorithm uses a combination of a rule-based and fuzzy logic classification schemes to achieve this end. The cloud types identified by the 2BCL4 product and their main defining characteristics are identified in Table 1. Factors taken into account in the classifier include cloud top and base height and temperature, as well as cloud phase, thickness, horizontal extent and cover. Different thresholds for cloud top/base heights are chosen for polar regions, tropics, and mid-latitudes.





**Table 1.** Cloud types identified by the 2BCL4 cloud classification algorithm and some of the properties upon which the algorithm is based. Abbreviations: cloud base (CB), horizontal extent (HE), vertical extent (VE), liquid water content (LWP). Adapted from Table 3 in Wang (2012).

| Cloud type | CB (km) | HE (km) | VE (km) | LWP ($kg.m^{-2}$) | Rain |
|---|---|---|---|---|---|
| High cloud (Ci) | 7– | 1–1000 | 1–7 | = 0 | none |
| Altostratus (As) | 2–7 | 1000 | 1–7 | ≈ 0 | none |
| Altocumulus (Ac) | 2–7 | 1000 | 0–7 | > 0 | virga possible |
| Stratus (St) | 0–2 | 100 | 0–1 | > 0 | none or slight |
| Stratocumulus (Sc) | 0–2 | 1000 | 0–1 | > 0 | drizzle or snow possible |
| Cumulus (Cu) | 0–3 | 1– | 0–7 | > 0 | drizzle or snow possible |
| Deep Convective (DC) | 0–3 | 10– | 7– | > 0 | intense shower of rain or hail possible |
| Nimbostratus (Ns) | 0–4 | 50–1000 | 7– | > 0 | prolonged rain or snow |

Reported cloud phase is restricted to ice for cloud base temperatures below −38.5 °C, and mixed-phase or liquid for temperatures above +1 °C. Although CALIOP provides the depolarization ratio to identify cloud phase, it is not reliable alone due to multiple scattering and the fact that the CALIOP

signal is quickly attenuated in multi-layer and thick clouds. Instead, it is used in combination with the attenuated backscatter coefficient and radar reflectivity, and exploits differences in the number concentration, vertical distribution, and radiative properties of ice particles and water droplets to distinguish different phase clouds when this cannot be uniquely determined by cloud top/base temperature alone. In this study, the stratus (St) and stratocumulus (Sc) cloud types have been agglomerated

based on advice released on the CloudSat website (CloudSat Data Processing Center, 2016).

The area of interest in this study covers as much of the RIS as possible and extends into the southern Ross Sea. Defined by the edges of the ice shelf (160° E to −150° E), it extends from the bottom of the A-Train track at 82° S north to 75° S. The area is divided into two sectors (RIS and the Ross Sea) along the 78° S circle of latitude, each of which are further divided into east/west sectors to form

four quadrants. Figure 1 identifies the study region and its bounds. Note that despite the larger area of the Ross Sea compared to the RIS defined in this study, the number of vertical profiles linked to the RIS region is far higher than that for the Ross Sea (4.1 million versus 1.8 million). This disparity is associated with a strong latitudinal variation in the sampling density associated with the satellite orbits. This study uses observations made between 1st January 2007 and 31st December 2010 when

both the CloudSat and CALIPSO satellites were fully operational and aligned.

In Section 3 we inspect cloud occurrence as a function of altitude for different cloud phases and the cloud fraction. The cloud occurrence is derived by counting the occurrence of cloud at a particular altitude using the CloudLayerBase and CloudLayerTop fields of the 2BCL4 product. This is in contrast to the methodology used in Verlinden et al. (2011), which used a threshold of 50%





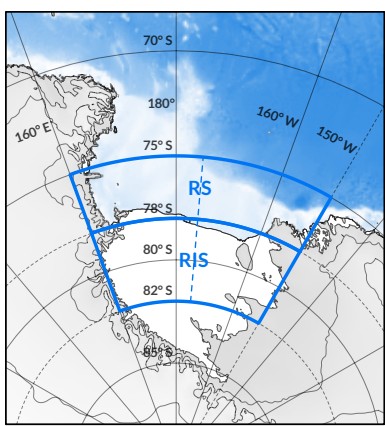

**Figure 1.** Topographic map of the RIS and Ross Sea including the boundaries of the study area (thick blue line) and border between the RIS and Ross Sea sectors. (Map derived from the SCAR Antarctic Digital Database.)

for the CloudFraction field (fraction of lidar volumes in a radar resolution volume that contains hydrometeors) for determination of cloud occurrence. In this study we calculate the cloud fraction (not related to the CloudFraction field) as the complement of the clear sky fraction, which is the number of clear sky profiles divided by the total number of profiles. As such, it is independent of altitude.

**2.2  Synoptic Climatology**

To provide context on atmospheric circulation over the duration of this study, classifications and regimes developed in the work of Coggins et al. (2014) and Coggins and McDonald (2015) are used. Five broad synoptic-scale regimes, hereafter referred to as "Coggins regimes", encompass 20 classes created by applying the *k*-means clustering technique to 10 m winds from 33 years of ERA-Interim

reanalysis (Dee et al., 2011) over the RIS/Ross Sea region. The 20 classes grouped into five regimes were found to be representative of conditions in the area and span the entire time period of available cloud observations so are an obvious choice for this analysis. The first two Coggins regimes are the weak northern cyclonic (WNC) and strong northern cyclonic (SNC) regimes which feature cyclones to the north of the RIS, with the "weak" and "strong" ratings referring to their effect on the winds

over the RIS; WNC generally provides weak forcing and low wind speeds while SNC features a strong synoptic pressure gradient force and high wind speeds over the RIS. The Ross Ice Shelf airstream (RAS) Coggins regime covers the strongest winds over the RIS and typically features a strong cyclone to the north and east that provides a large pressure gradient over the ice shelf which forms RAS-like signatures (Parish et al., 2006), while the weak southern cyclonic (WSC) regime is



**Table 2.** Relative frequency of occurrence of the Coggins regimes annually (all) and seasonally (DJF–SON) in the ERA-Interim reanalysis (%). DJF/MAM/JJA/SON correspond to austral summer/autumn/winter/spring respectively. Values for seasons are normalized so that rows sum to 100% (not including "all").

|      | all | DJF | MAM | JJA | SON |
|------|-----|-----|-----|-----|-----|
| **WNC** | 23 | 24 | 34 | 19 | 23 |
| **SNC** | 14 | 22 | 27 | 26 | 25 |
| **RAS** | 25 | 10 | 28 | 37 | 25 |
| **WSC** | 9  | 12 | 36 | 30 | 23 |
| **WS**  | 29 | 36 | 17 | 21 | 27 |

associated with relatively weak cyclones and mesocyclones positioned over the RIS with medium wind speeds. Finally, the weak synoptic (WS) Coggins regime covers periods where a very weak pressure gradient and very low winds are present over the RIS.

Table 2 shows the relative frequency of occurrence of the regimes over the entire observational period examined and a normalized seasonal frequency. Examination of the "all" column shows that the WSC regime is relatively rare (9% annual frequency) while the WS regime is observed frequently (29%). The WNC and RAS regimes are also quite common (25% and 23% respectively) while SNC is less common (14%). Seasonal analysis of the frequency of occurrence shows the WNC regime occurs most frequently during austral autumn (34%) but much less frequently during winter (19%), while the SNC regime is more uniform across all four seasons – this likely reflects the ubiquitous nature of synoptic-scale cyclonic storms around Antarctica (Hoskins and Hodges, 2005). The RAS regime is seen much more frequently in winter (37%) than summer (10%), while the WSC and WS regimes alternately favour autumn (WSC 36%) and summer (WS 36%) at the expense of summer (WSC 12%) and autumn (WS 17%). It must be noted that Table 2 is structured for seasonal analysis of individual regimes (rows sum to 100%) and does not provide a comparable statistic of regime frequency in each season (season columns do not sum to 100%).

## 3   Results

As an initial point of comparison with the previous Antarctic-wide studies of Verlinden et al. (2011) and Adhikari et al. (2012) we display seasonal mean cloud occurrence statistics in Figure 2 derived from the 2BGL4, 2BGL5 and the 2BCL4 products for the Ross Sea and the RIS. Comparison between the three products is generally rather good, though the 2BGL4 values of cloud occurrence are a little above the values derived from the other two products everywhere below 8 km. A step change in the cloud occurrence can also be observed at 8.2 km in all seasons and over both the Ross Sea and the RIS in the 2BGL4 product. It should be noted that this step change is particularly large in the winter and spring and is also significantly larger over the RIS than the Ross Sea. The 2BGL5 and





2BCL4 values do not display this discontinuity and are much more similar to each other, though it is
noticeable that the 2BCL4 values of cloud occurrence are always smaller than the other two products
below 1 km. Interestingly the temporal average cloud occurrence for the 2BGL4 product is always
larger than that for the 2BCL4 product, which in turn is always greater than the 2BGL5 product.

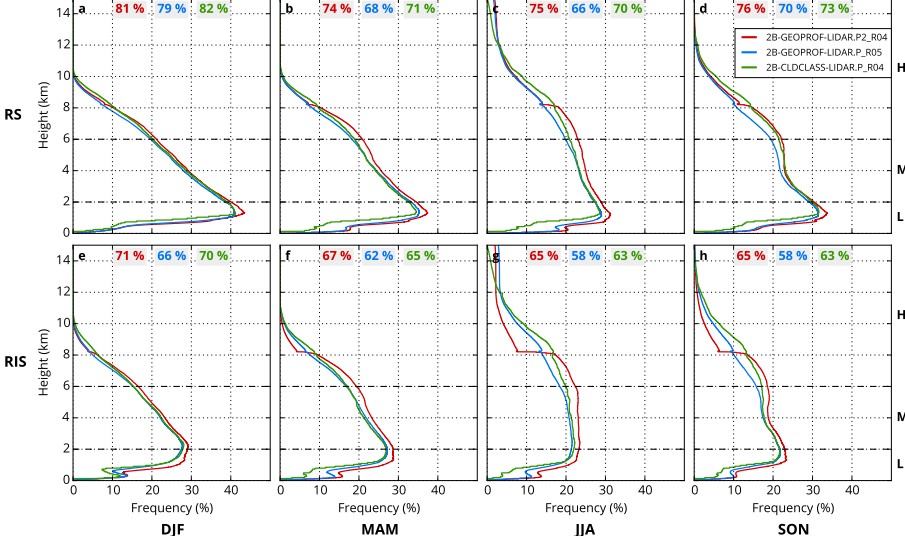

**Figure 2.** Mean vertical profiles of cloud occurrence derived from the 2BGL4, 2BGL5 and 2BCL4 data for the
Ross Sea (a–d) and RIS sectors (e–h) for different seasons. Total sector cloud fraction (temporal average cloud
occurrence independent of altitude) is annotated at the top of each sub-figure. L, M, and H labels indicate the
low, medium, and high cloud regions, respectively, as discussed in the text.

To further examine the extent of this issue, Figure 3 displays the zonal mean value of the ratio of
the cloud occurrence at 8.3 km to the cloud occurrence at 8.0 km derived from the three products.
The two altitude bins are in consecutive height bins, but are linked to different vertical and horizontal
resolutions in the 2BGL4 processing scheme according Mace et al. (2009). Inspection of Figure 3
shows that the ratio varies between 0.9 and 1.1 for all three products near the equator and at mid-
latitudes. However, the 2BGL4 value of the ratio deviates significantly from that derived from the
other two products at latitudes poleward of 75° in both hemispheres. The deviation between the
2BGL5 product and the 2BCL4 product is also relatively large in the Northern Hemisphere above
60° N. Previous studies (Verlinden et al., 2011; Bromwich et al., 2012; Adhikari et al., 2012) have
highlighted this discontinuity near 8 km, but have questioned whether it is an instrumental artefact
or a physical feature. The analysis in Figure 3 clearly suggests that this is an instrumental artefact
specific to both polar regions. The larger discontinuity observed in Figure 2 in winter may suggest a





temperature dependent issue. But, further analysis is beyond the scope of this study given the good correspondence between the 2BCL4 and 2BGL5 products.

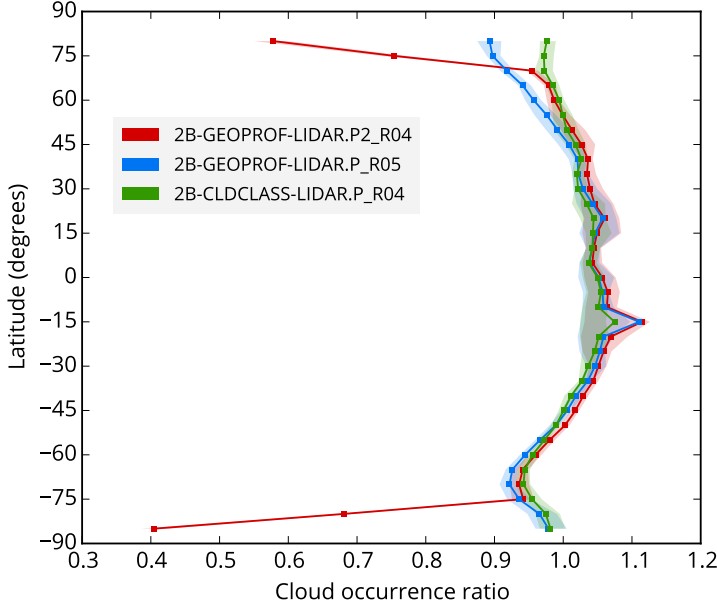

**Figure 3.** Latitudinal variation of the ratio of cloud occurrence at 8.3 km to cloud occurrence at 8.0 km derived from the 2B-GEOPROF-LIDAR products and the 2B-CLDCLASS-LIDAR products. The envelopes represent the interquartile ranges of the ratio observed at that latitude. Note that consistently anomalous values $\ll 1$ are confined to the polar latitudes.

Given the uncertainty identified within the 2BGL4 (GEOPROF) product we choose to deviate from previous studies and use the 2BCL4 (CLD-CLASS) product. We consider this preferable to the 2BGL5 product, despite the apparent resolution of the uncertainties in 2BGL4, as it provides information on cloud phase and type which are particularly interesting in this region. We initially examine cloud occurrence as a function of cloud phase. Adhikari et al. (2012) reported that the largest seasonal variations in cloud occurrence were observed over the RIS and sea ice region in the surrounding Ross Sea, suggesting this region may be of particular interest in understanding the controls of cloud in the region. Cloud occurrence is separated into two sectors: the Ross Sea and the RIS (see Figure 1). For the purpose of this analysis we separate clouds into three vertical ranges: low-level clouds (0–2 km), mid-level clouds (2–6 km) and high-level clouds (6– km) identified by horizontal lines in Figures 2, 4 and 5.

Figure 4 (a–d) displays the cloud occurrence for the Ross Sea region in each season broken into different cloud phases (shaded areas). The maximum cloud fraction (82%) is observed during sum-





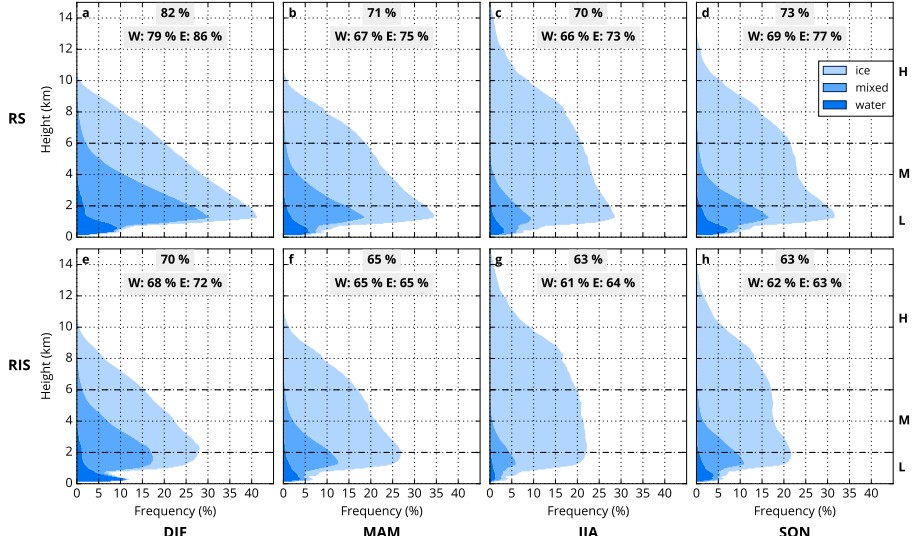

**Figure 4.** Mean vertical profiles of cumulative cloud occurrence for different cloud phases derived from 2BCL4 data for the Ross Sea (a–d) and RIS sectors (e–h) for different seasons. Total sector cloud fraction (temporal average cloud occurrence independent of altitude) is annotated at the top of each sub-figure, along with values for the western (W) and eastern (E) halves of each sector. L, M, and H labels indicate the low, medium, and high cloud regions, respectively, as discussed in the text.

mer, with the minimum cloud fraction (70%) observed during winter. The largest cloud fraction is observed over the eastern portion of the Ross Sea in every season with the greatest seasonal cloud fraction in summer (86%). The smallest cloud fraction (66%) was observed in winter over the western portion of the Ross Sea. Cloud occurrence as a function of altitude shows the same pattern with the maximum (about 40%) occurring in summer and the minimum (about 27%) during winter. Though all maxima occur between 1.5 and 2 km above sea level (ASL), the winter maximum is noticeably weaker. Mixed phase clouds are predominant near the cloud occurrence altitudinal maximum, with water and ice cloud contributing roughly equally to the remainder. Changes in the occurrence of mixed phase cloud appear to constitute the majority of the change in the cloud occurrence at that altitude.

Cloud occurrence reduces uniformly at increasing altitude from the maxima in summer and autumn, while in winter the cloud occurrence reduces rapidly between the peak and 3.5 km ASL then remains relatively uniform, before a more rapid reduction at higher altitudes (8 km in winter and 6 km in spring). Previous work detailed in Verlinden et al. (2011) highlighted a discontinuity in cloudiness near 8 km ASL over much of the continent which appears to be linked to the processing artefact identified previously.





.

Unlike the study of Verlinden et al. (2011) we do not observe two distinct maxima in the vertical profiles of the cloud occurrence, however this feature was relatively weak over the WAIS (see Figure 5 in Verlinden et al., 2011) which may hint at the specific drivers of the cloud environment in this region. In particular, the absence of a secondary peak in mid- and high-level cloud occurrence is interesting given the ubiquitous nature of cyclones in the region (Hoskins and Hodges, 2005) and the muted seasonal signal in cloud occurrence above 2 km ASL could be explained by the lack of a strong seasonal signal in cyclone frequency in this region (supported by the small seasonal signal in the frequency of the SNC regime displayed in Table 2). The difference in our cloud occurrence calculation methodology to that used by Verlinden et al. (2011) may have some impact, however it is unlikely to explain all of the difference.

As might be expected, the liquid water phase occurs predominantly in low-level clouds with a local maximum between 300 and 900 m ASL in all seasons, the largest enhancement occurring during summer. The difficulty of detecting cloud within 1 km of the ground using CloudSat due to ground clutter (Mace et al., 2009; Haynes et al., 2011) may bias low-level cloud detection in favour of periods of reduced attenuation of the CALIOP lidar instrument (clear sky or optically thin mid- to high-level cloud). Mid-level (between 2 and 6 km ASL) cloud occurrence varies little between seasons at the upper limit (6 km) at close to 20%, but mid-level cloud fraction is greatest in summer and lowest in winter. More high-level cloud (above 8 km ASL) is observed during winter than summer which matches with the results identified in Adhikari et al. (2012). We also note that clouds were not observed in this study above 10 km ASL in both the summer and autumn, or 12 km in spring, but were seen above 14 km in the winter. Haynes et al. (2011) suggestion that the maximum cloud height over the Southern Ocean will be impacted by the seasonal variations in tropopause depth likely explains this pattern, which interestingly shows a similar seasonal progression to that for polar stratospheric cloud occurrence (Alexander et al., 2011, 2013).

The mean seasonal cloud occurrence vertical profiles for the RIS are displayed in Figure 4 (e–h). The eastern portion of the RIS has slightly greater cloud occurrence than the western portion of the RIS in all seasons apart from autumn. The cloud fraction for the RIS area is 70% in summer and between 63 and 65% in all other seasons. The greatest cloud fraction by sector is observed over the eastern RIS during summer (72%), while the lowest is observed in winter over the western RIS (61%). Inspection of the cloud occurrence as a function of altitude shows a maximum at approximately 2 km ASL in every season, slightly higher than the altitude of the peak observed over the Ross Sea. This peak in cloud occurrence matches with a similar peak in ice water content (IWC) and liquid water content (LWC) values discussed in Scott and Lubin (2016) over Ross Island (located at the south-west corner of the Ross Sea region in this study). Summer experiences the greatest cloud occurrence as a function of altitude at this peak, but the seasonal variability is rather muted (just under 5% variation across all seasons) relative to that observed over the Ross Sea (approximately




12%), with the minimum in winter. The mixed phase class is a contributor to this peak in all seasons, but is dominant in summer. The cloud occurrence linked to the ice phase is slightly larger than that for mixed phase cloud in autumn.

Again, the water phase occurs predominantly for low-level clouds (below 2 km ASL) with maxima below 600 m observed in every season (this is particularly clear for summer). The ice phase is effectively the only contributor for high-level clouds (above 6 km) and is the largest contributor to cloud occurrence in every season, though mixed phase cloud is dominant up to approximately 3 km in summer. The quantity of mixed and water phase cloud as a proportion of total cloud occurrence is substantially lower than that observed over the Ross Sea, possibly suggesting a lack of moisture in this region and the impact of colder temperatures.

The seasonal variation in cloud occurrence at mid-levels (between 2 to 6 km ASL) is approximately 5–10%, which is smaller than the variations observed at the peak occurrence level. Within this altitude range, cloud occurrence is more constant in winter and spring and reduces with altitude in spring and summer. The cumulative occurrence of mid-level clouds is marginally higher in summer than other seasons, with a minimum value in winter. High-level clouds (above 6 km) are distinctly more common in autumn and winter than spring and summer. Similar to the Ross Sea case, the majority of clouds are limited to below 10 km ASL in both the summer and autumn, with maximum high-cloud occurrence in winter. Over the entire vertical profile, the seasonal variation over the RIS is smaller than that observed over the Ross Sea (c.f. Figure 4 (a–d) and (e–h)).

We now examine composites of the vertical distribution of cloud occurrence and cloud fraction for these two regions based on the synoptic-scale Coggins regimes. Figure 5 (a–e) displays the cloud occurrence profiles over the Ross Sea for the 5 different Coggins regimes. Comparison of Figure 5 (a–e) shows that the combined cloud occurrence (associated with the three different cloud phase classes) in every regime again maximizes just below 2 km ASL. The greatest occurrence at that peak is observed in the SNC regime and the smallest occurrence in the WSC regime. At middle altitudes (4 to 6 km), the difference in cloud occurrence between regimes is very large with the SNC regime again displaying the largest cloud occurrences (between 28 and 35% in this altitude range) and the WSC regime the least (between 7 and 17%). The WSC and WS regimes have rather similar vertical profiles with the vast majority of cloud occurrence linked to low- to mid-level clouds below 4 km ASL. The SNC type on the other hand has high cloud occurrence at nearly all levels compared to the other regimes and has signs of a secondary maxima at 5 km. The RAS and WNC have intermediate levels of cloud occurrence between the SNC and WSC/WS regimes, with shallower rates of reduction in the cloud occurrence above the peak.

Inspection of the distribution of phases linked to the SNC regime (linked to strong cyclonic activity in the north of the Ross Sea) suggests that this regime is dominated by ice cloud at all levels down to about 2 km ASL (i.e. mid- to high-level clouds are predominately comprised of ice). A small amount of water phase cloud is observed very close to the surface and a peak in mixed phase cloud is


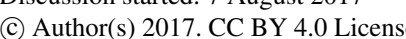

observed just below the peak in the combined cloud occurrence. The quantity of mixed phase cloud being larger than the ice phase only occurs in the WS and WNC types near the low-level maxima. The SNC and RAS regimes feature the largest proportion of ice cloud overall. The cloud fraction (see labels in Figure 5) varies from 82% for the SNC regime to 58% for the WSC regime, with

the other regimes having values between 72 and 77%. This variation is significantly larger than that observed when observations are composited based on season. This result suggests that cloud fraction is strongly impacted by synoptic situation. In particular, the SNC regime has high occurrence frequencies in the mid- to high-level cloud region above 2 km ASL. Additionally, we note that the variation between the western and eastern portions of the Ross Sea is larger in the SNC, RAS, and

WSC regimes (10–13%) than over the seasons (7–8%), while the WNC regime shows little variation longitudinally. This suggests that synoptic forcing is a more important control on longitudinal differences than season, though this should be expected because the strength and position of cyclonic centres is a principal determinant of the Coggins regimes. While overall the synoptic typing seems to be important, we note that the proportion of liquid and mixed phase cloud varies relatively little

over the Ross Sea as a function of the Coggins regimes (between approximately 15 and 20% at the altitude of maximum occurrence), while the seasonal variation is substantially larger (between approximately 9 and 30% at the altitude of maximum occurrence). This result supports the view that temperature is a strong driver of the occurrence of ice cloud as previously identified by Haynes et al. (2011), though the variability observed between synoptic types suggests that temperature anomalies

associated with specific synoptic types also have some influence.

Figure 5 (f–j) displays composites for the RIS for the Coggins regimes. Cloud occurrence is significantly smaller in every regime relative to the profiles over the Ross Sea (c.f. Figure 5 (a–e)). Examination of the cloud occurrence profiles as a function of altitude for each regime suggests that the WNC, WSC, and WS regimes have similar forms, as do the SNC and RAS regimes (to

each other). Interestingly, cloud occurrence is higher in the RAS regime between 2 and 6 km than SNC. This likely suggests that the impact of cyclones in the northern Ross Sea is not as strong an influence on cloud over the RIS. At mid- to high-levels (above 4 km), RAS and the SNC have the largest cloud occurrences with ice cloud dominating in this region. The variation at upper levels is also noticeably larger between the various synoptic regimes in Figure 5 than between the seasons

displayed in Figure 4. This seems to suggest that the synoptic state is a stronger driver of mid- to high-level cloud than seasonal variations, this being particularly clear when we consider that the regime with most high-level cloud (the SNC regime) displays almost no seasonality (see Table 2). Figure 5 therefore shows an advantage in using a classification scheme based on synoptic states relative to one using seasons in this region. Previous work by Haynes et al. (2011) also suggested

that seasonality might not be a strong influence over the Southern Ocean with two exceptions, these being the quantity of ice cloud and the height at which the maximum cloud fraction occurs in the upper troposphere.





Examination of the cloud fraction over the entire RIS and the western and eastern sectors shows less variability than over the Ross Sea. The cloud fraction varies only between 55 and 68% and the
differences in cloud fraction between the western and eastern sectors is only sizeable (9%) for the WSC regime. The difference between the cloud fraction between the western and eastern sectors is 5% or less in all other regimes. Given that the WNC and SNC regimes are dominated by the positions of cyclones over the Ross Sea this may not be surprising in those cases. However, the lack of longitudinal variation associated with the RAS which is traditionally linked to flow near the TAM
is a surprise.

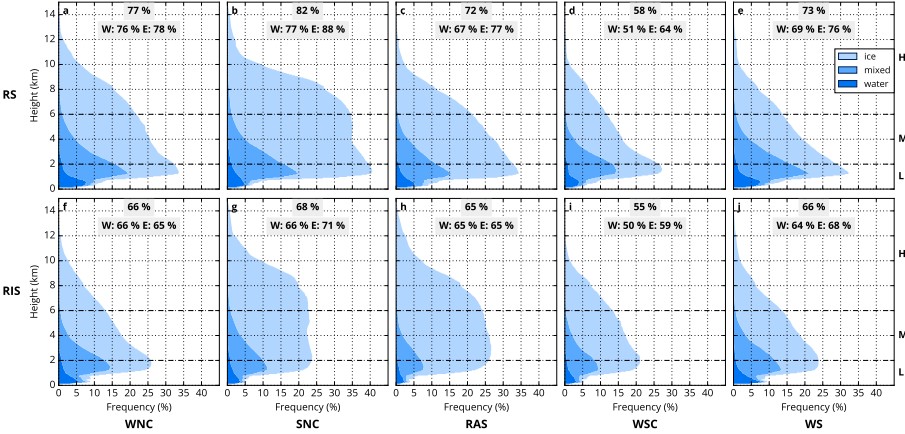

**Figure 5.** Mean vertical profiles of cumulative cloud occurrence for different cloud phases derived from 2BCL4 data for the Ross Sea (a–e) and RIS regions (f–j) for the Coggins regimes. L, M, and H labels indicate the low, medium, and high cloud regions, respectively, as discussed in the text.

While useful, the mean vertical profiles of cloud occurrence displayed in Figures 4 and 5 do not fully represent the individual profiles composited in that season or regime. For example, two states associated with a distinct high and low cloud type might be combined in the averaging process to form the mean cloud occurrence observed. Alternatively, multi-layered cloud might be present and
contribute to the mean cloud occurrence profiles. In an effort to display this variability, Figure 6 shows the quantity of clear skies, single-layer cloud, and multi-layer cloud for the Coggins regimes and seasons over the Ross Sea and RIS. Inspection of Figure 6 (a) for the Ross Sea region suggests that clear skies are observed 26% of the time on average. However, when the cloud occurrence information is composited based on Coggins regime the frequency of occurrence of clear skies varies
between 18% for the SNC regime to 42% for the WSC regime. Seasonal variations in clear sky occurrence are considerably smaller at 18 to 30%. Again, this highlights that clouds are observed preferentially in the Ross Sea when strong cyclonic centres are observed in the northern Ross Sea. Changes in the frequency of multi-layer clouds are also notable with the frequency varying from



15% for the WSC regime to 33% for the SNC regime. The occurrence of multi-layer cloud is con-
siderably more constant as a function of season, varying between 21 and 25%, which highlights that
the quantity of multi-layer cloud is also strongly impacted by synoptic conditions.

Figure 6 (b) displays the occurrence of clear skies, single layer cloud and multi-layer cloud for the
RIS region. The variability as a function of both Coggins regime and season is again muted relative
to the Ross Sea region. The occurrence of clear skies varies from 32% for the SNC regime to 45%
for WSC, with the other three regimes having frequencies between 34 and 35% which is similar to
that for the SNC regime. This suggests that only the synoptic conditions linked to the WSC regime
are strongly linked to clear skies. When clear sky occurrence is examined as a function of season
a very small seasonal variability is observed (values fall between 30 and 37%). This reinforces our
previous conclusion for the Ross Sea region: that clear skies are not strongly influenced by season
and therefore surface temperatures. Examination of multi-layer cloud values shows a variation be-
tween 15% for WSC and 23% for both the RAS and SNC types. This suggests that the RAS regime
is also preferentially related to multi-layer cloud over the RIS. Work by Steinhoff et al. (2009) has
previously suggested that the RAS regime might be linked to the occurrence of low- and mid-level
cloud, the latter being associated with vertical ascent generated by low-level convergence as the RAS
decelerates downstream of wind speed maximum along the TAM. This relationship also appears to
be observed based on our statistical analysis. The high occurrence of cloud at mid-levels (between
2 and 6 km ASL) displayed in Figure 2 (g) therefore suggests that the RAS and possibly the marine
air intrusions identified in Nicolas and Bromwich (2011) have a noticeable climatological impact on
cloud occurrence over the RIS. In addition, the seasonal progression and variation linked to synoptic
typing display very different impacts over the Ross Sea and RIS. This perhaps highlights the stronger
influence of cyclones on cloud occurrence over the Ross Sea than the RIS. The higher frequency of
occurrence of multi-layer cloud linked to the SNC type over the Ross Sea than the RIS also suggests
the position of the cyclone centre plays an important role in cloud distributions.

Figure 7 displays the fractional occurrence of the various cloud types over the Ross Sea and RIS
composited based on Coggins regime and season. For the sake of conciseness, we will only discuss
the types which have substantial fractional occurrence rates (above 15% in any class). The frequency
of nimbostratus (Nb) is so small over these regions that this type is not included in Figure 7. For the
Ross Sea, the most commonly occurring cloud type is deep convective (DC) which varies from 32%
for the WS regime to 43% for the RAS regime, with the other regimes ranging between 32 and
39% (see Figure 6 (a)). The seasonal variation has a maximum in autumn of 42% with a minimum
of 36% in winter. Thus, in this high-level cloud type, more variation between classes is associated
with synoptic classification than season. Interestingly the maximum occurrence of this type over the
Ross Sea is associated with the RAS regime rather than the SNC regime, previously identified as the
regime associated with the highest cloud occurrence. When we additionally include the impact of
clear skies (values displayed in Figure 6) this conclusion remains unchanged.



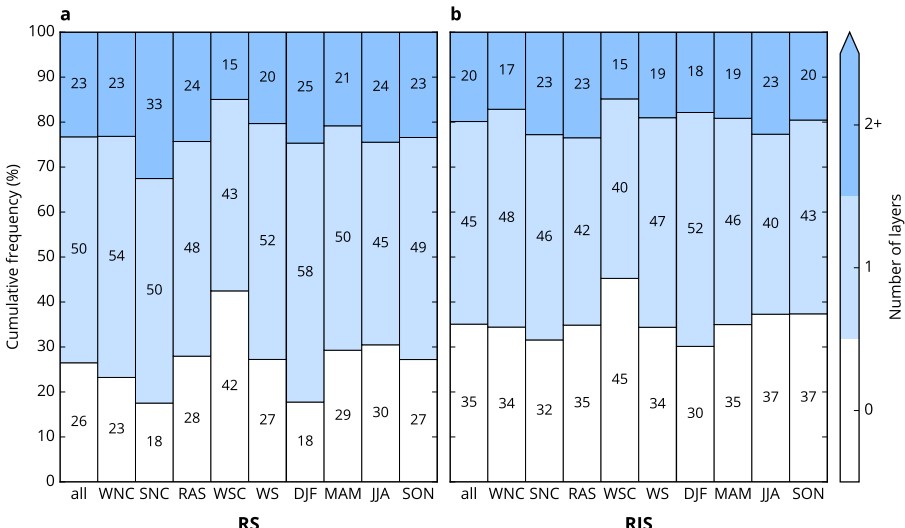

**Figure 6.** Distribution of the number of cloud layers over the Ross Sea and RIS for all cases, the Coggins regimes and season.

The next most common cloud type over the Ross Sea is the altostratus (As) type, which varies between 30 and 36% based on Coggins regimes and between 29 and 36% based on season. In particular, the fractional occurrence of this cloud type maximizes in winter and minimizes in summer over the Ross Sea. However, when the frequency of clear skies (see Figure 6) is also considered, the

seasonal variation becomes very small (23 to 25%), while the regime variation is enhanced to 17 to 30%. Thus, in this case close inspection also suggests that synoptic forcing is a driver of the occurrence of this cloud type with the highest occurrence in the SNC regime and lowest occurrence in the WSC regime. Note that the As type is predominantly a mid- to high-level cloud dominated by ice and thus may not be strongly impacted by seasonally varying quantities, such as sea ice cover

and surface temperature.

Low-level clouds (combined stratus/stratocumulus or St/Sc types) are observed relatively frequently in the WS (21%) and WSC (19%) regimes over the Ross Sea. Both these regimes are associated with weaker synoptic forcing and observed less frequently than the regimes with stronger synoptic forcing. Seasonal variation in these types changes from 17% in summer to 12% in winter.

Thus, it seems that the occurrence of this class is more associated with periods of weak synoptic forcing, note that these conditions occur more often in summer (see Table 2), which in turn might suggest that local factors are important. Inclusion of information on clear sky rates does not change this result.




Figure 7(b) displays cloud type fractional frequency information for the RIS region. Over the RIS, the As cloud type is most prevalent varying between 38 and 46% based on synoptic regime and 25 and 43% based on season. However, when clear sky occurrence is considered these values reduce to 24 to 32% for the regimes and 25 to 27% for the seasons. This suggests that the quantity of altostratus remains nearly constant seasonally. The highest occurrence of the As type when clear skies are taken into consideration are linked to the SNC and RAS regimes, with very similar low occurrences (24 to 25%) for the other regimes.

The next most prevalent cloud type over the RIS is the DC type, which changes between occurrence rates of 23 and 33% for the various Coggins regimes and 24 and 30% based on seasons. The highest fractional occurrence of the DC type occurs for the RAS regime and the minimum is, surprisingly, linked to the SNC type. Thus, two regimes which are related to strong synoptic forcing in the region have very different impacts on this cloud type. This latter result might be explained by the position of the cyclonic centres, preferentially in the north east Ross Sea for the SNC type, relative to the RIS. When the frequency of clear skies is included in our analysis, a larger variation in this cloud type is linked to synoptic forcing than seasonal changes.

The combined St and Sc cloud types also have an appreciable occurrence rate over the RIS (13%). When the Coggins regimes are considered this type has a minimum occurrence of 7% linked to the RAS regime and a maximum occurrence of 20% linked to the WNC regime. It should be noted that there is an obvious change in the fractional frequency of this type between strong synoptic forcing regimes (RAS and SNC) and weaker synoptic forcing regimes (WNC, WSC, and WS). This separation again suggests that these clouds are linked to periods of weak synoptic forcing. The range of the fractional occurrence rates associated with the different seasons is again smaller than that associated with the synoptic types; summer displays the highest occurrence rate. When clear sky frequencies are included in calculations this result is unchanged.

The fraction of the cirrus (Ci) cloud type is also appreciable over the RIS and has the same frequency of fractional occurrence as the combined St and Sc cloud type (13%). The fraction of this cloud type maximizes at 17% for the SNC regime and has a minimum occurrence of 9% for the WNC regime. The Ci type is observed most frequently in winter (20%) and least in summer (6%). Thus, synoptic variations do not appear to be a very strong control on this cloud type. This conclusion is unchanged when the occurrence of clear skies is included in our analysis. We note that the larger fractional occurrence of Ci over the RIS compared to the Ross Sea could be associated with the proximity of the TAM to the RIS and the influence of isolated cirrus generated by orographically-forced waves, this conjecture being supported by Haynes et al. (2011) and Scott and Lubin (2016).

Figure 8 (a) and (f) display joint histograms of the cloud top height versus the geometric cloud thickness over the Ross Sea and RIS, respectively. Figure 8 (a) shows that low-level (below 2 km ASL) thin cloud (thicknesses below 3 km) has a high occurrence over the Ross Sea, an observation previously identified by Adhikari et al. (2012). However, thin clouds are observed relatively





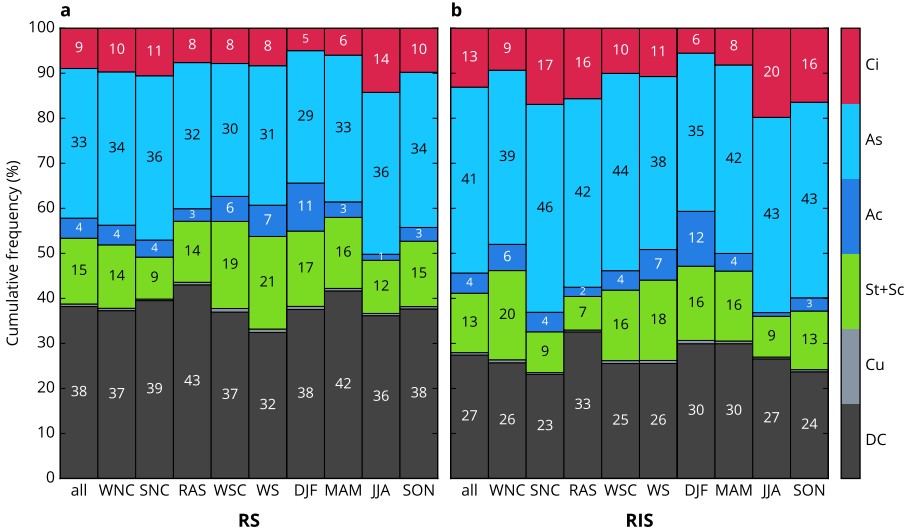

**Figure 7.** Percent fraction of cloud types over the Ross Sea and RIS for all cases, the Coggins regimes and seasons. The cloud types are identified in Table 1.

frequently for cloud top heights between the surface and approximately 8 km over the Ross Sea. Clouds that effectively cover nearly the complete vertical column to cloud top (i.e. that have similar thicknesses to their cloud top height) are also observed frequently. We also note that clouds with high cloud tops (above 6 km) are relatively rare. The logarithmic scale associated with Figure 8 highlights

that thin low-level cloud is very common. A similar pattern is observed over the RIS region overall (see Figure 8 (f)), though cloud occurrence is higher in general for the Ross Sea, particularly at lower levels. This is to be expected, given the differences in solar heating of the surface, sea ice concentration changes and sea surface temperatures. In particular, the greater availability of moisture over the Ross Sea associated with open water in summer would likely be an important contributor. The

overall pattern is similar to the joint histograms identified by Haynes et al. (2011) over the Southern Ocean.

To further understand the distribution of clouds over the two regions, Figure 8 (b–e) and (g–j) displays anomalies from the annual means for each season for the Ross Sea and RIS, respectively. Examination of Figure 8 (b–e) suggests that the anomaly patterns are rather similar in the summer

and autumn with higher cloud occurrence observed for low-level (below 2 km) and mid-level (2 to 6 km) cloud which covers the majority of the vertical column up to the cloud top height. Lower cloud occurrence is associated with thin high top clouds for summer and autumn. The anomaly patterns in winter and spring are near mirror images of those in summer and autumn with higher cloud occurrence with thin high top cloud and lower occurrence (relative to the annual mean) for low-level



and mid-level cloud covering the vertical column. High top clouds (above 8 km) with a range of
thicknesses are also enhanced in winter and spring, with the enhancement being more noticeable
for thick clouds in winter. Haynes et al. (2011) has previously identified that the increase in thicker
clouds in winter over the Southern Ocean may be associated with storm track activity. Haynes et al.
(2011) also suggests that the maximum cloud height might vary seasonally based on the tropopause
height, therefore this also seems like a reasonable explanation for the enhanced occurrence of high-
level cloud above 8 km in the winter relative to the summer. The anomaly patterns for each season
are generally rather similar over the Ross Sea and the RIS. This is surprising since this may imply
that moisture availability is not a large driver of cloud.

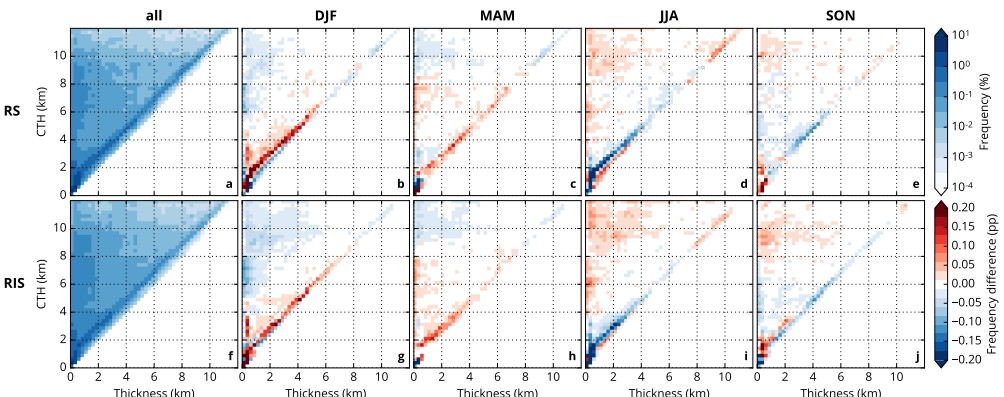

**Figure 8.** Joint histogram of the cloud top height vs. geometric cloud thickness over the Ross Sea and RIS for
the entire year on a logarithmic scale (a, f) and the difference from the annual mean over the respective region
(RS and RIS) for each season on a linear scale (b–e, g–j).

Figure 9 (b–f) and (h–l) display the anomalies from the mean associated with the Coggins regimes
for the Ross Sea and RIS, respectively. Figure 9 (a) and (g) display histograms of the frequency of
occurrence as a function of thickness and cloud top height for the Ross Sea and RIS, respectively, and
are exact reproductions of Figure 8 (a) and (f) and are included to aid in interpretation. Inspection of
the anomalies from the mean linked to different synoptic regimes shows some interesting structure.
The SNC regime is linked to a dearth of low-level cloud relative to the mean, particularly over the
Ross Sea, while the WSC regime is linked to a considerable enhancement in the frequency of low-
and mid-level cloud which covers the majority of the vertical column up to the cloud top height.
These variations may also explain the small quantity of multi-layer cloud in this regime over the
Ross Sea. The enhancement in the WSC regime is also observed in the WS regime, but that regime
is also related to a stronger reduction in clouds covering the vertical column above 4 km. For the SNC
regime, the dearth in low-level cloud over the Ross Sea is counter-balanced by an increase in thick
(greater than 7 km deep) high-level cloud relative to the mean, which also accounts for the increased




cumulative cloud occurrence for the SNC regime identified in Figure 5 (b) and Figure 6 (a). While
some aspects of the anomaly for the RAS regime (see Figure 9 (d) ) are similar to the SNC regime
pattern (see Figure 9 (c)), notably a reduction in low-level cloud relative to the mean, differences

can be observed. For example, the clouds with high cloud tops (above 8 km ASL) are under- rather
than over-represented relative to the mean for the entire thickness range for RAS compared to SNC.
Enhanced cloud occurrence in the RAS regime is primarily limited to mid-level clouds, most notably
linked to clouds with thicknesses below 2 km and the region identifying that the cloud covers nearly
the full atmospheric column.

To put the anomaly patterns identified for the Ross Sea into context, it is also worthwhile consid-
ering the patterns over the RIS (Figure 9 (h–l)). Unlike the seasonal analysis presented in Figure 8,
which displayed large similarities for the anomaly patterns over the Ross Sea and RIS, the patterns
show more variability for the WNC, SNC, and RAS regimes between the two regions. In particular,
the WNC regime displays a strong enhancement in the quantity of low- and mid-level cloud below

4 km ASL, with thin cloud and cloud covering the majority of the atmospheric column up to the
cloud top height being enhanced. The SNC type shows a similar pattern to that over the Ross Sea,
but the strong enhancement of deep, high cloud top height, cloud is not observed in this case. The
RAS regime joint histogram shows a stronger reduction in low- and mid-level cloud below 4 km
over the RIS than the Ross Sea and the enhanced cloud occurrence region now occurs for high-level

cloud (cloud top height above 6 km ASL). Thus, the vertical extent of the RAS regime change con-
siderably between the Ross Sea and the RIS. Based on previous work detailed in Steinhoff et al.
(2009) this may be associated with vertical ascent over the RIS.

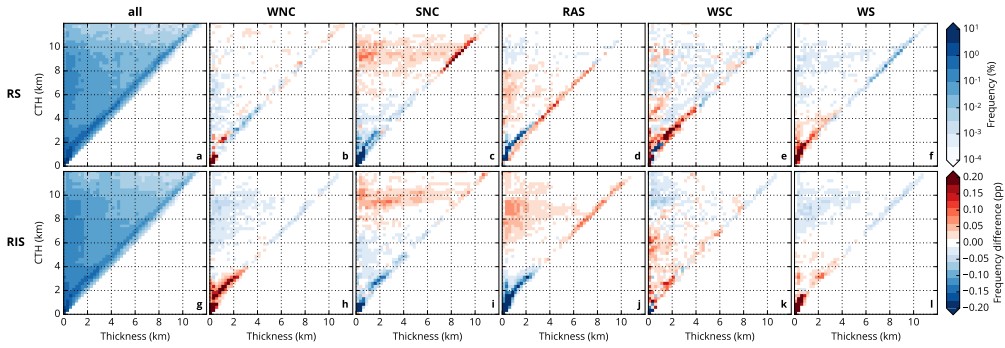

**Figure 9.** Joint histogram of the cloud top height vs. geometric cloud thickness over the Ross Sea and RIS for
the entire year on a logarithmic scale (a, f) and the difference from the annual mean over the respective region
(RS and RIS) for each Coggins regime on a linear scale (b–e, g–j).





## 4   Conclusions and Discussion

This study has quantified the vertical distribution of cloud fraction, phase, and type over the Ross
Ice Shelf and southern Ross Sea using four years of data from the 2B-CLDCLASS-LIDAR R04
product (Sassen et al., 2008) composited using seasons and synoptic regimes (Coggins and McDonald, 2015). The following results highlight the usefulness of incorporating a synoptic classification
scheme in the climatological analysis of clouds in this region:

- Large differences exist between the cloud occurrence as a function of altitude for synoptic
regimes relative to those for seasonal variation (c.f. Figures 4 and 5).

- There is strong variation in clear sky and multi-layer cloud occurrence as a function of synoptic
      regime as opposed to season (see Figure 6).

- There is higher variance in all cloud type occurrences, apart from the cirrus type, over both the
      Ross Sea and RIS associated with synoptic type compared to seasonal composites (see Fig-
ure 7) which remains true when the frequency of clear skies is taken into account or discounted
      from our analysis.

- Anomalies from the mean joint histogram of cloud top height against thickness display sig-
      nificant differences over the Ross Sea and RIS sectors as a function of synoptic regime, but
      are near identical over these two regions when a seasonal analysis is completed (see Figures 8
and 9).

- Clouds are observed preferentially in the Ross Sea when strong cyclonic centres are observed
      in the northern Ross Sea.

- The cumulative cloud occurrence observed in the western and eastern portions of the Ross Sea
      and RIS display larger differences for composites based on the synoptic regimes than seasons.
This again suggests a significant influence of the position of cyclonic centres. However, the
      the lack of longitudinal variation associated with the RAS which is traditionally linked to flow
      near the TAM is unexpected.

We have therefore proven that an analysis based on synoptic regimes explains more of the variation in overall cloud occurrence and specific cloud types than a simple seasonal analysis. This com-
plements previous studies which have inferred these relationships (Verlinden et al., 2011; Adhikari
et al., 2012) or used a case study approach (Steinhoff et al., 2009; Scott and Lubin, 2014). It is how-
ever important that the seasonal component of this analysis is not disregarded; it can more effectively
capture variations in temperature and subsequently moisture availability via both the water holding
capacity of the air and the presence/absence of open ocean due to seasonal sea ice. For example,
seasonal analysis identified that the occurrence of mixed phase and liquid water cloud varies more

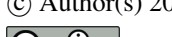



strongly as a function of season than regime, suggesting seasonal variability in mean temperature is a strong driver of ice cloud as previously identified by Haynes et al. (2011).

We also examined the 2BGL4 and 2BGL5 data products. The 2BGL4, used in previous studies in this region, displays a discontinuity at 8.2 km which is not observable in the other products and
appears to correspond with a change in the horizontal and vertical resolution of the CALIPSO dataset used above this level (Mace et al., 2009). This discontinuity appears to occur at latitudes poleward of 75° in both hemispheres.The 2BGL5 product appears to have addressed this issue.

Figures 4 and 5 identify that cloud occurrence as a function of altitude is dominated by low-level cloud, peak cloud occurrence occurring below 2 km in every season and synoptic regime. This
supports previous work by Haynes et al. (2011) and Adhikari et al. (2012) which indicated that there is a relatively high occurrence of low-level cloud above the Southern Ocean and Antarctica, respectively. Adhikari et al. (2012) also suggested that low-level cloud constitute the major cloud type in Antarctica and are more frequent during summer than winter. Our analysis also suggests that stratus and stratocumulus are more common in summer than winter (see Figure 7) over both the Ross
Sea and RIS. Separation into different synoptic classes also implies that periods of weak synoptic forcing (WNC, WSC, and WS Coggins regimes) are important for the formation of these clouds. The greater prevalence of these types over the Ross Sea and RIS also suggests that sea ice state and temperature could be important factors.

Adhikari et al. (2012) also identified that high-level and deep clouds are more frequent in winter
and spring than summer. The deep convective (DC) cloud type is observed to have a maximum in winter and spring and lowest occurrence rates in summer consistent with the result indicated in Adhikari et al. (2012). However, examination of the variations in the frequency of this cloud type with synoptic regime also suggest that this is most often observed during periods linked to strong cyclonic activity (the SNC regime) as hypothesized by Adhikari et al. (2012). Our synoptic
classification additionally identifies that the cloud fraction appears to largely be controlled by the SNC regime which is linked to strong cyclones in the northern Ross Sea, however RAS events also seem to be a strong controlling factor during winter over the RIS.

The results of this synoptic classification also strongly support the representative nature of the case studies detailed in Scott and Lubin (2014) which identified significantly contrasting cloud properties
above Ross Island associated with different meteorological regimes. For example, they identified that warm, moist air moving directly over Ross Island from the north brought low clouds which were likely predominantly liquid phase. Our analysis shows that there is far more liquid water cloud (and also mixed phase cloud) over the Ross Sea than the RIS in every season and for every synoptic type. Thus, any southward flow is likely to have this impact.
In contrast, clouds within marine air masses arriving from the WAIS, and descending onto the Ross Ice Shelf before reaching Ross Island, show strong ice phase signatures based on the study of Scott and Lubin (2014). Our analysis also shows that the RAS regime displays large quantities of



ice cloud at all levels over the RIS. The SNC regime is also predominately linked to ice clouds at all levels down to about 2 km ASL. The fact that the SNC and RAS regimes were dominated by ice

phase cloud is likely associated with the strong vertical motions linked to these synoptic types. This result is inferred from the discussion in Scott and Lubin (2016) which identified that cloud ice water content is strongly impacted by vertical motion.

The highest cloud occurrence was found over the eastern Ross Sea quadrant during the summer, while the lowest cloud occurrence is observed over the western halves of both the RIS and Ross

Sea sectors during the winter. We observe a link between strong synoptic forcing (as judged by wind speeds over the RIS and Ross Sea) and greater occurrence of high-level cloud (above 6 km ASL), while regimes linked to reduced synoptic forcing seem to be related to a greater occurrence of low-level cloud.

The strong changes in cloud occurrence vertical distribution, cloud fraction and cloud type as-

sociated with specific synoptic types allows us to make some wider inferences based on analysis of the Coggins regimes. For example, Coggins and McDonald (2015) demonstrated that the depth and location of the Amundsen Sea Low have significant impacts over the Ross Sea and RIS. Thus, we can infer that changes in the depth of the Amundsen Sea Low will likely have caused significant changes in the cloud environment over the Ross Sea and RIS. The variability in cloud types

for different synoptic conditions and the importance of some types for precipitation also suggests that changes in synoptic forcing over the region related to the Amundsen Sea Low may well have impacted snow accumulation in the region. In particular, the high frequency of occurrence of the DC cloud type, a type linked to intense precipitation events statistically (see Table 1), during the RAS regime suggests that snow accumulation in this region may be strongly modulated by the occurrence

rate of this synoptic regime. This will be an area of further work.

*Acknowledgements.* The authors acknowledge the ECMWF for the ERA-Interim dataset. The CloudSat/CALIPSO data were obtained from the CloudSat Data Processing Center. The SCAR Antarctic Digital Database was obtained from the British Antarctic Survey Geodata Portal.



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
