# Peer review of "An analysis of the cloud environment over the Ross Sea and Ross Ice Shelf using CloudSat/CALIPSO satellite observations: The importance of synoptic forcing"

_Atmospheric Chemistry and Physics, 2017_

## Referee Comment (RC1) · Anonymous Referee #1 · 29 Nov 2017

This is a very detailed description of clouds over the Ross Sea and Ross Ice Shelf based on analysis of a combined CloudSat and CALIPSO data set as a function of synoptic conditions and seasonality. The work is competently done. The results are of climatological interest to those specialists dealing with the atmospheric processes in this region. Readability will be greatly enhanced by adding subsections to Section 3. This will help to keep one's attention focused on the detailed analyses being presented. I would add the specific labels to Figure 2: 2BGL4, 2BGL5, and and 2BCL4.

---

## Referee Comment (RC2) · Anonymous Referee #3 · 17 Feb 2018

A comprehensive study of the clouds over the Ross Ice Shelf and the Ross Sea is presented. Clouds are analyzed for different seasons and classified into types in order to identify relationships with synoptic fields of temperature, humidity etc. The manuscript shows some interesting results and can be published after minor revisions.

Line 162: I am a bit puzzled by the statement "Reported cloud phase is restricted to ice for cloud base temperatures below $-38.5$ C, and mixedphase or liquid for temperatures above $+1$ C". I would have expected mixed phase clouds between $-38.5°C$ and $0°C$ and liquid clouds above $0°C$. Could you clarify?

[Figure]

Line 225: "Comparison between the three products is generally rather good" - this is a subjective statement which should be replaced by something quantitative.

Line 438: I wouldn't have expected deep convection to be the predominant cloud type in the Arctic. Maybe this is simply a misnaming for a vertically extended cloud?
* * *

---

## Author Comment (AC1) · 31 Mar 2018

**Response to Anonymous Referee #1**

**We would like to thank the reviewer for their effort. Responses to the reviewer's comments are identified below in BOLD and reference changes we have made to an updated version of the manuscript.**

This is a very detailed description of clouds over the Ross Sea and Ross Ice Shelf based on analysis of a combined CloudSat and CALIPSO data set as a function of synoptic conditions and seasonality. The work is competently done. The results are of climatological interest to those specialists dealing with the atmospheric processes in this region.

Readability will be greatly enhanced by adding subsections to Section 3.
This will help to keep one's attention focused on the detailed analyses being presented.

**We believe that this is a good suggestion to help break up the text, we have added the following section headings in an updated manuscript**

**3.1 Discontinuity in 2B-GEOPROF-LIDAR R04 Product**
**3.2 Cloud Occurrence and Phase by Season**
**3.3 Cloud Occurrence and Phase by Synoptic Regime**
**3.4 Multilayer Cloud by Season and Regime**
**3.5 Cloud Type by Season and Regime**
**3.6 Cloud Height and Thickness**

I would add the specific labels to Figure 2: 2BGL4, 2BGL5, and and 2BCL4.

**We believe the reviewer is asking us to change the labels in the key of the Figure 2 from 2B-GEOPRFO_LIDAR P2_R04 to 2BGL04 etc. We believe that this improves clarity and is thus a change we have made in a revised manuscript. For consistency we have also made this change in Figure 3.**

---

## Author Comment (AC2) · 31 Mar 2018

Response to Anonymous Referee #3

**We would like to thank the reviewers for their effort. Responses to reviewers comments are identified below in BOLD and reference changes we have made to an updated version of the manuscript.**

A comprehensive study of the clouds over the Ross Ice Shelf and the Ross Sea is presented. Clouds are analyzed for different seasons and classified into types in order to identify relationships with synoptic fields of temperature, humidity etc. The manuscript shows some interesting results and can be published after minor revisions.

Line 162: I am a bit puzzled by the statement "Reported cloud phase is restricted to ice for cloud base temperatures below − 38.5 C, and mixed phase or liquid for temperatures above +1 C". I would have expected mixed phase clouds between -38.5∘C and 0∘C and liquid clouds above 0∘C. Could you clarify?

**We would like to thank the reviewer for identifying this error. We believe that replacing the original sentence with the following sentence rectifies this issue.**

**"Reported cloud phase is restricted by cloud base and cloud top temperature. For cloud base temperature below − 38.5 °C, only ice cloud is permitted. For cloud base temperature between -38.5 and 1 °C, all phases are permitted (liquid, ice and mixed). For cloud base temperature above 1 C, the cloud is classified as liquid when cloud top temperature is above -7 °C, liquid or mixed when the cloud top temperature is between -38.5 and -7 °C or mixed for cloud top temperature below -38.5 C (Wang et al., 2012)."**

**Reference:**
**Wang, Z., Vane, D., Stephens. G, and Reinke, D.(2012), Level 2 Combined Radar and Lidar Cloud Scenario Classification Product Process Description and Interface Control Document, Version 1, JPL document.**

Line 225: "Comparison between the three products is generally rather good" - this is a subjective statement which should be replaced by something quantitative.

**We agree that this is a subjective statement, but it was meant as a high level generalisation, we suggest the refined wording below would improve this:**

**"A visual comparison between the three products generally shows good agreement (within a few percentage points)."**

Line 438: I wouldn't have expected deep convection to be the predominant cloud type in the Arctic. Maybe this is simply a misnaming for a vertically extended cloud?

We use the Deep Convection identification used in the 2BCL4 dataset. We agree that this may not be the best description in the Antarctic, but feel it is important to remain consistent with the dataset definitions. We have added the line:

"This cloud type is likely identified due to large horizontal and vertical extent of the cloud rather than presence of deep convection in the polar region."